# Critical Requirement of SOS1 for Development of BCR/ABL-Driven Chronic Myelogenous Leukemia

**DOI:** 10.3390/cancers14163893

**Published:** 2022-08-11

**Authors:** Carmela Gómez, Rósula Garcia-Navas, Fernando C. Baltanás, Rocío Fuentes-Mateos, Alberto Fernández-Medarde, Nuria Calzada, Eugenio Santos

**Affiliations:** 1Centro de Investigación del Cáncer, Instituto de Biología Molecular y Celular del Cáncer, CSIC-University of Salamanca and CIBERONC, 37007 Salamanca, Spain or; 2Instituto de Biomedicina de Sevilla (IBiS), Hospital Universitario Virgen del Rocío, CSIC, Universidad de Sevilla, 41013 Seville, Spain; 3Departamento de Fisiología Médica y Biofísica, Universidad de Sevilla, 41013 Seville, Spain

**Keywords:** SOS1, SOS2, RAS, RAC, GEF, CML, myeloproliferative disorders, SOS-KO phenotypes

## Abstract

**Simple Summary:**

The p210^BCR/ABL^ oncoprotein is necessary and sufficient to trigger chronic myelogenous leukemia (CML) in mice. Our prior in vitro studies showing that the ABL-mediated phosphorylation of SOS1 promotes RAC activation and contributes to BCR-ABL leukemogenesis suggested the significant role of SOS1 in the development of CML. To provide direct in vivo experimental evidence of the specific contribution of SOS1 to the development of CML, here, we analyzed the effect of the direct genetic ablation of SOS1 or SOS2 on the genesis of p210^BCR/ABL^ -driven CML in mice. Our data showed that direct SOS1 genetic ablation causes the significant suppression of all the pathological hallmarks typical of CML, demonstrating that SOS1 deficiency is protective against CML development and identifying this cellular GEF as a relevant, novel therapeutic target for the clinical treatment of this hematological malignancy.

**Abstract:**

We showed previously that the ABL-mediated phosphorylation of SOS1 promotes RAC activation and contributes to BCR-ABL leukemogenesis, suggesting the relevant role of SOS1 in the pathogenesis of CML. To try and obtain direct experimental evidence of the specific mechanistic implication of SOS1 in CML development, here, we combined a murine model of CML driven by a p210^BCR/ABL^ transgene with our tamoxifen-inducible SOS1/2-KO system in order to investigate the phenotypic impact of the direct genetic ablation of SOS1 or SOS2 on the pathogenesis of CML. Our observations showed that, in contrast to control animals expressing normal levels of SOS1 and SOS2 or to single SOS2-KO mice, p210^BCR/ABL^ transgenic mice devoid of SOS1 presented significantly extended survival curves and also displayed an almost complete disappearance of the typical hematological alterations and splenomegaly constituting the hallmarks of CML. SOS1 ablation also resulted in a specific reduction in the proliferation and the total number of colony-forming units arising from the population of bone marrow stem/progenitor cells from p210^BCR/ABL^ transgenic mice. The specific blockade of CML development caused by SOS1 ablation in p210^BCR/ABL^ mice indicates that SOS1 is critically required for CML pathogenesis and supports the consideration of this cellular GEF as a novel, alternative bona fide therapeutic target for CML treatment in the clinic.

## 1. Introduction

The foremost hallmark of chronic myelogenous leukemia (CML), the Philadelphia (Ph) chromosome, arises from a t (9;22) translocation fusing the Abelson murine leukemia (ABL1) locus on chromosome 9 with the breakpoint cluster region (BCR) gene on chromosome 22 [1]. This translocation gives rise to chimeric fusion proteins, collectively named BCR-ABL1, whose constitutively activated tyrosine kinase activity is ultimately responsible for the altered differentiation, uncontrolled replication, and resistance to apoptosis typically displayed by CML cells. Most CML patients present a BCR-ABL1 fusion downstream of exons 13 and 14 of the BCR gene, originating transcripts with an e14 and/or an e13 junction, which drive the expression of a 210 kDa chimeric protein designated p210 BCR-ABL1 (M-BCR), although other isoform fusion proteins (namely p190 and p230) have also been described at lower frequencies in some patient subtypes [2,3,4,5,6]. The enduring proliferation of CML stem cells expressing BCR-ABL1 fusion proteins eventually potentiates the occurrence of additional mutations, which are often associated with a more negative prognosis and the eventual development of resistance to clinical treatments [7,8].

Whereas it is clearly apparent that the hyperactive, uncontrolled kinase activity of the chimeric fusion BCR-ABL proteins is responsible for their oncogenic potential and the development of CML [9,10], the various downstream signaling pathways mediating their transforming ability are not yet fully defined [11,12]. Among these, various reports have shown that the pathway of activation of RAC GTPases is critically required for BCR-ABL-mediated malignant transformation [13,14,15,16]. In this regard, using CML cells and mouse xenografts, we recently showed that the phosphorylation of SOS1 on tyrosine 1196 promotes its RAC GEF activity and contributes to BCR-ABL leukemogenesis [17], indicating that the BCR-ABL-pY1196SOS1-RAC axis is essential in promoting the full transformation and leukemogenic activity of the fusion protein.

The ubiquitously expressed members of the SOS family (SOS1, SOS2) are the most universal and functionally relevant GEFs capable of activating RAS and RAC GTPases (and, consequently, their specific downstream signaling cascades) in a variety of biological contexts [18,19,20,21,22,23]. Whereas SOS1 is essential for embryonic development [24], SOS2 is dispensable to reach adulthood in mice [25]. We managed to bypass the embryonic lethality of SOS1-KO alleles by using a floxed, tamoxifen-inducible SOS1 null mutation that allowed the generation and parallel analysis of the adult, viable mice of four relevant SOS genotypes (WT, SOS1-KO, SOS2-KO, and SOS1/2-DKO). Using this genetic KO system, we recently demonstrated the functional prevalence of SOS1 over SOS2 regarding the control of MEF cellular proliferation and viability, as well as a direct mechanistic link between SOS1 and the control of intracellular mitochondrial redox homeostasis [19,26], and we also characterized the specific functional roles of SOS1 and SOS2 in different biological contexts [27,28,29,30].

Regarding tumoral pathogenesis, our earlier studies of SOS1- and/or SOS2-knockout mice also suggested the critical contribution/participation of SOS1 in the development of RAS-dependent, DMBA/TPA chemically induced skin tumors [31] and Bcr/Abl-driven chronic myeloid leukemia (CML) [17]. In particular, we showed that SOS1 is required for the BCR/ABL-mediated activation of RAC, cell proliferation, and transformation, in vitro in K562 cells and in vivo in a xenograft mouse model. We observed the significantly delayed development of leukemia upon the transplantation of bone marrow-derived cells (BMDC) expressing a retrovirally inserted p210^BCR/ABL^ construct into irradiated recipient mice devoid of SOS1. Since, under physiological conditions, SOS1 is known to be expressed in myeloid cells and is regulated by the ABL kinase in the processes of RTK-induced actin cytoskeleton remodeling [32], our observations in SOS1/2-KO mice suggest that SOS1 may be a critical mechanistic contributor to the pathogenesis of CML and thus may constitute a valuable potential therapeutic target for the treatment of this disease.

To obtain further direct experimental evidence supporting these hypotheses, here, we wished to evaluate the effect of SOS1 and/or SOS2 genetic ablation in a mouse model of p210^BCR/ABL^-driven CML and derived cell lines. For this purpose, we recently generated an ad hoc mouse colony by cross-mating a transgenic p210^BCR/ABL^-driven CML mouse model [33,34] with our SOS1/2-KO system, allowing us to generate WT, SOS1-KO, SOS2-KO, and SOS1/2-DKO mice [26,35]. In this combined {SOS1/2KO | Tec-BCR-ABL^p210^} genetic system, we investigated, in detail, the phenotypic effects of direct SOS1 and/or SOS2 depletion on the initiation/progression of CML in adult mice. Our observations in this experimental system showed that SOS1 genetic ablation causes a specific suppression of CML development in this mouse model, thus validating the consideration of SOS GEFs as new potential therapeutic targets for this disease that may be particularly useful in those cases where resistance to traditional TKI treatment arises.

## 2. Materials and Methods

### 2.1. Mouse Models for CML Analysis

The combined {Tec-p210^BCR/ABL^|SOS1/2-KO} mouse model used in this report was generated by performing relevant cross-matings (Figure 1A) between a transgenic mouse strain, kindly provided by Dr. Honda [33], and mice of appropriate genotypes from our TAM-inducible SOS1/2-KO colony [19,26,31,35]. As previously described [35] a mouse strain harboring a floxed version of SOS1 with exon 10, flanked by LoxP sites (Sos1^fl/fl^), was crossed with SOS2-KO mice [24,25] in order to generate our conditional SOS1/2-KO mouse model, where SOS1 ablation is controlled by CreERT2. All animals in the colony were kept on the same C57BL/6J background and maintained and treated under identical experimental conditions.

Because TAM is an antagonist of the estrogen receptor, a 10-day washout habituation period using a soy-free diet (Teklad 16% global protein rodent diet, catalog no. 2916; Harlan) was given to the animals prior to the actual TAM treatments. Afterward, a TAM-containing chow diet (Harlan; Teklad CRD TAM400/CreER) was administered, and experimental determinations were performed on specific sets of control and KO animals, as indicated. TAM treatment for a short period (4 days) triggered visible systemic, whole-body Cre recombinase activation, and the progressive removal of the floxed SOS1 gene product was complete after 12 days of treatment, as detected by PCR assays with specific primers or WB assays using specific antibodies (Figure 1B,C).

Mice were kept, managed, and sacrificed in the NUCLEUS animal facility of the University of Salamanca according to current European (2007/526/CE) and Spanish (RD 1201/2005 and RD53/2013) legislation. All experiments were approved by the Bioethics Committee of the Cancer Research Center (#415).

### 2.2. Peripheral Blood Analysis

Blood samples were taken from the submandibular sinus at 7 months of age into microvette ethylenediaminetetraacetic acid (EDTA)-coated tubes (Sarstedt Inc., Nümbrecht, Germany) to monitor hematological parameters and disease progression. To analyze these samples, HEMAVET 950 (Drew Scientific Inc., Miami Lakes, FL, USA) was used, and the following parameters were monitored: hemoglobin (Hb), total white blood cells (WBC), red blood cells (RBC), platelets (PLT), percentage of neutrophils (NE), lymphocytes (LY), monocytes (MO), eosinophils (EO), and basophils (BA).

### 2.3. Histological Analyses

For histological examination, the tissues of interest were dissected, fixed for 24 h with 4% paraformaldehyde, and washed in phosphate buffer. The processing and staining of the sections were performed by the PMC-BEOCyL Unit (Comparative Molecular Pathology-Biobank Network of Oncological Diseases of Castilla y León). In brief, tissue blocks were then dehydrated and paraffin-embedded. Paraffin sections (3 μm thick) were mounted on Superfrost-Plus slides (Thermo Scientific, Limburg, Germany) and deparaffinized with xylene, followed by decreasing concentrations of ethanol and distilled water. Hematoxylin–eosin (H&E) staining was performed using standard procedures.

### 2.4. Isolation and Purification of Hematopoietic Stem Progenitor Cells (HSPCs)

Mice were sacrificed by cervical dislocation. The femur was removed, and the bone marrow was rinsed. The filtrate was centrifuged, and the pellet was suspended in red blood cell lysis buffer at room temperature, centrifuged at 3000 rpm, 5 min. Bone marrow mononuclear cells (MNCs) were suspended in PBS containing EDTA and 0.5% BSA, and hematopoietic stem cells/progenitor cells (HSPCs) were isolated by using the CD117 [36,37] MicroBead Kit (Miltenyi Biotec, Bergisch Gladbach, Germany) immunomagnetic beads sorting technique with lineage cell depletion kits and anti-c-Kit microbeads, according to the manufacturer’s instructions.

### 2.5. Colony-Forming Unit (CFU) Assays

The HSPCs were incubated overnight in RPMI media supplemented with 10% FBS and a cytokine cocktail with FLT3 ligand (20 ng/mL), IL-6 (10 ng/mL), TPO (10 ng/mL), and mSCF (50 ng/mL). After 12 h of stimulation, the cells were plated on MethoCult GF M3434 media (STEMCELL Technologies, Vancouver, BC, Canada) containing 3 μM imatinib or 2 μM 4-hydroxy tamoxifen (alone or in combination) on three replicate plates. Colony numbers were recorded after 1 week of plating. MethoCult™ media (M3434) support optimal growth of burst-forming unit-erythroid (BFU-E), granulocyte, and/or macrophage progenitor cells (CFU-granulocyte, macrophage (CFU-GM)), as well as multi-potential progenitor cells (CFU-granulocyte, erythrocyte, macrophage, megakaryocyte (CFU-GEMM)) [38,39]. The classes of mouse hematopoietic progenitor cells detected using MethoCult™ were counted after 10–14 days and identified by their size and morphology, as described below. BFU-E contains groups of tiny and irregular cells that were identified at high magnification. CFU-GM includes granulocyte and macrophage; these colonies contain multiple cell clusters with dense cores. Macrophage lineage cells are large cells with an oval-to-round shape and appear to have a grainy or gray center. The granulocytic lineage cells are round, bright, smaller, and more uniform in size than macrophage cells. CFU-GEMM produces large colonies with >500 cells containing cells of at least two lineages. These colonies have a dense core with an indistinct border between the core and peripheral cells.

### 2.6. Pan-Cancer Analysis of Gene Dependency Scores in the DepMap Portal

DepMap identifies cancer vulnerabilities by systematically identifying genetic and pharmacologic dependencies and the biomarkers that predict them in different types of human tumors [40]. The hSOS1 and hSOS2 gene dependency scores were analyzed here in the current datasets of DepMap 22Q2 Public + Score of the DepMap web portal (https://depmap.org/portal/, accessed on 22 June 2022). A lower dependency score indicates a higher likelihood that the gene of interest is essential for a given tumor cell line. A 0 indicates the gene is not essential and a −1 is comparable to the median of all pan-essential genes.

### 2.7. Statistical Analysis

GraphPad Prism 8.0.1 (GraphPad Inc., San Diego, CA, USA) software was used. All data presented are an average of at least four independent experiments performed in triplicates. Results are expressed as mean ± S.E.M. Differences between experimental groups were analyzed using one-way ANOVA and Bonferroni’s tests. No statistical method was used for the predetermination of sample size; animal numbers were minimized to conform to ethical guidelines while accurately measuring parameters of animal physiology.

## 3. Results

### 3.1. SOS1 Deficiency Protects from Death in a Murine Model of p210^BCR/ABL^ Induced CML

To make it possible to carry out a direct experimental test of the in vivo impact of the ablation of SOS1 and SOS2 GEFs on the development of CML in mice, we generated a mouse colony resulting from cross-mating a transgenic strain expressing p210^BCR/ABL^ under the control of the promoter of the mouse Tec gene [33,34], which starts developing full-blown CML symptoms between 5 and 8 months of age with our TAM-inducible SOS1/2-KO system, allowing us to produce the individual or combined ablation of SOS1 and/or SOS2 in mice [26,35] (Figure 1). These crosses gave rise to mice with different genotypic combinations that, upon appropriate TAM treatment for SOS1 ablation, allowed us to monitor and compare the evolution of p210^BCR/ABL^-induced CML in the littermates of appropriate and relevant SOS genotypes, including WT, SOS1-KO, SOS2-KO, or SOS1/2-DKO mice (Figure 1).

The TAM treatment (to induce SOS1 ablation) of appropriate strains from our p210^BCR/ABL^|SOS1/2-KO mouse colony allowed us to generate SOS1-KO, SOS2-KO, and WT adult, transgenic mice, which were used to evaluate the effect of single SOS1 or SOS2 genetic ablation on the lifespan and survival curves of mice suffering CML caused by their endogenously expressed p210^BCR/ABL^ transgene (Figure 2). Unfortunately, we were unable to carry out similar CML survival analyses in mice lacking both SOS1/2 isoforms since SOS1/2-DKO mice die precipitously after about two weeks of the concomitant, full-body ablation of both SOS1/2 GEFs [35].

Consistent with the original description of the p210^BCR/ABL^ transgenic mouse strain used to generate the mouse colony used in these studies [33,34], our control, the WT transgenic mice, started showing visible CML-related hematological alterations around 6–8 months of age, and in the ensuing months, these alterations quickly progressed toward the massive hematological alterations typical of full-blown CML that eventually cause the death of these animals.

Interestingly, the comparison of the Kaplan–Meier survival curves of WT, SOS1-KO, or SOS2-KO transgenic mice expressing p210^BCR/ABL^ showed that the single ablation of SOS1 or SOS2 extended the lifespan and survival of the CML-diseased transgenic mice in comparison with WT transgenic animals expressing normal levels of both GEFs (Figure 2). Importantly, in the analysis of mice cohorts of the appropriate genotypes and controls grown under similar conditions in parallel, SOS1 appeared to be significantly more relevant than SOS2 regarding CML development since SOS1 ablation resulted in about a ~4-month increase in CML survival and a 40% increase in lifespan compared to the WT controls (Figure 2A) in comparison with the ~2-month increase in survival shown by the SOS2-KO mice (Figure 2C).

It is also relevant to mention that the ablation of SOS1 only produced a significant delay with respect to CML when SOS1 was ablated in the early stages of CML disease development (6 months of age) (Figure 2A). In contrast, the TAM-induced ablation of SOS1 after 8 months of age, when CML was already significantly developed in the transgenic mice, did not produce any significant delay in the disease (Figure 2B). Consistent with this, significant statistical differences were also computed (*p* < 0.01) if we directly compared the survival profiles of SOS1-KO mice treated with TAM after 6 months of age (Figure 3A) or after 8 months of age (Figure 3B).

Furthermore, it is also worth noting that the SOS1-dependent delay of CML observed here is not an off-target effect of TAM treatment since TAM treatment did not by itself cause any changes in the kinetics of the survival of the control p210^BCR/ABL^ transgenic mice that were also WT for both SOS1 and SOS2. In the non-transgenic model, single SOS1 or SOS2 disruption did not affect mice survival either (Figure 2 and Appendix A).

### 3.2. Reversal of Altered CML Hematological Parameters after SOS1 (But Not SOS2) Ablation in p210^BCR/ABL^ Transgenic Mice

As previously described, the initial CML-related hematological alterations in the p210^BCR/ABL^ transgenic mouse strain used in our studies started to be easily detectable at about 6 months of age and then quickly progressed during the ensuing 2–4 months (7–9 months of age) until culminating into the full-blown disease and the eventual death of the diseased animals. To ascertain the mechanistic relevance of the SOS1 and SOS 2 GEFs in the development of CML, here, we compared the various hematological parameters of the WT, SOS1-KO, and SOS2-KO transgenic (p210^BCR/ABL^) and non-transgenic mice (Figure 3).

Consistent with previous reports [33,34], our analysis of the temporal evolution of several typical CML-related hematological alterations showed that our p210^BCR/ABL^ transgenic mouse strains conserved the normal expression of SOS1 and SOS2 (WT for SOS1/2) and displayed a quick increase in the overall count of WBC and platelets, as well as a decrease in hemoglobin (HB) levels that was also concomitant with a marked increase in the relative population of granulocytic cells (neutrophils) and a decrease in the lymphocyte population in the peripheral blood of these animals (Figure 3A upper, black). Interestingly, the SOS2-KO transgenic mice showed very similar profiles to those of equally aged WT mice with regard to the temporal changes produced in all those hematological parameters (Figure 3A upper, blue). In contrast, the profiles of the SOS1-KO transgenic mice of the same age showed an almost complete abolition of the development of all the pathological alterations typical of CML and displayed rather normal levels of WBC, HB, and platelets or the relative percentage of the different granulocytic and lymphocytic subpopulations (Figure 3A red, upper). Importantly, this effect of SOS1 ablation, abolishing the development of CML-linked hematological alterations, is clearly specific since the analysis of non-transgenic mice of the same ages and SOS1/2 genotypes showed similar unchanged profiles for all these parameters in WT SOS1-KO and SOS2-KO mice (Figure 3B, lower).

Indeed, a pie chart representation of the percentage distribution of the different granulocytic and lymphocytic populations present in the peripheral blood of 7-month-old and 9-month-old animals (Figure 3C) provides clear visual evidence of the dramatic increase in the ratio of neutrophils/lymphocytes (blue/orange sectors, Figure 3C) that is a typical hallmark of CML progression, which was almost completely reverted in the SOS1-KO mice in comparison with their counterparts, openly diseased WT and SOS2-KO p210^BCR/ABL^ transgenic mice (Figure 3C).

Since non-transgenic mice of the three relevant SOS genotypes (WT, SOS1-KO, and SOS2-KO) tested at different ages always showed a similar percentage distribution in all these blood cell populations (Figure 3D), our data clearly indicate that the functional contribution of SOS1 (but not SOS2) is critically required for the development of CML in p210^BCR/ABL^ transgenic mice.

### 3.3. Loss of SOS1 or SOS2 Reduces Splenomegaly and Hepatomegaly and Restores Spleen Homeostasis in p210^BCR/ABL^ Mice

Splenomegaly and, less frequently, hepatomegaly are also often described as pathological alterations associated with CML [41,42]; thus, we also wished to test and compare the effect of SOS1/2 genetic ablation on the size and histological structure of these hematopoietic organs in our non-transgenic and transgenic p210^BCR/ABL^ mice (Figure 4).

As expected, the non-transgenic animals (WT, SOS1-KO, SOS2-KO) did not develop CML under any circumstances and did not show any alterations in these organs under our experimental conditions. On the other hand, consistent with our prior observations [17], our analysis of p210^BCR/ABL^ transgenic mice expressing normal levels of SOS1 and SOS2 (WT for both SOS1 and SOS2) showed that the development of p210^BCR/ABL^-driven CML in those animals was also accompanied by a significant increase in the relative weights of the spleen and the liver (black bars), as well as by important histological disorganization of the spleen (but not the liver) of the same diseased animals (Figure 4A,B).

Interestingly, we observed that the relative weight of both hematopoietic organs in p210^BCR/ABL^ transgenic animals devoid of SOS1 (after the tamoxifen treatment of transgenic SOS1^fl/fl^ mice) or SOS2 (constitutive SOS2−KO mice) remained very similar to that of the control non-diseased mice of the relevant SOS genotypes (Figure 4). As TAM treatment by itself did not cause any changes in the non-transgenic mice of any genotype (Figure 4), these data indicate that the reduction in organ weight (relative to WT) is specifically due to the absence of either SOS1 or SOS2 from the p210^BCR/ABL^ transgenic mice (Figure 4).

We also noticed that the structural histological organization of the spleen was significantly altered as a consequence of the CML development in our control transgenic mice expressing normal levels of both SOS1 and SOS2 (WT (both −TAM and +TAM) and SOS1^FL/fl^ kept in the absence of TAM (Figure 4), which displayed significantly more disorganized and less defined red pulp and white pulp areas within this hematopoietic organ. In contrast, this structural disorganization was very much reduced or inexistent in the spleens of mice devoid of SOS1 or SOS2, suggesting the role these GEFs play in the structural disorganization of the spleens of CML-diseased animals (Figure 4). In contrast, despite the differences in overall organ weight, no structural disorganization in the liver tissue was observed in comparisons between the WT and the SOS1-KO or SOS2-KO mice (Figure 4).

An analysis of a smaller number of older animals eventually reaching 14 months of age showed that these older SOS1/2-KO mice develop CML-related splenomegaly similar to that shown in young 9-month-old WT mice (Appendix A), suggesting that the SOS2-KO mice eventually develop similar, though later, CML phenotypes than WT mice carrying a normal dosage of SOS proteins. However, we also noticed that the 14-month-old SOS1-KO transgenic mice presented a clearly healthier profile than SOS2-KO mice of the same age with regard to the development of known CML markers, including splenomegaly and the NEU/LYMP ratio (Appendix A), thus confirming the dominant role of SOS1 over SOS2 with regard to CML pathogenesis. 

### 3.4. Effect of SOS1/2 Genetic Ablation on the Hematopoietic Stem Cell Population of p210^BCR/ABL^ Transgenic Mice

Our prior report on the significant temporal delay in the development of CML upon the transplantation of BMDC cells from SOS1-KO mice that were retrovirally infected with a p210^BCR/ABL^ construct into irradiated recipient mice [17] provided indirect evidence suggesting the participation of SOS1 in the process of CML development. In this regard, direct experimental evidence that SOS1 is critically required for CML development is now provided in this report by our in vivo observations, documenting that the genetic ablation of SOS1 (and SOS2, to a lesser extent) results in significant protection from CML development driven by p210^BCR/ABL^ in p210 transgenic mice (Figure 2, Figure 3 and Figure 4).

Hematopoietic stem cells (HSPCs) are relevant for CML pathogenesis and also for intrinsic or acquired therapy resistance, relapse, and disease progression [43,44]. In order to obtain a deeper mechanistic insight into the functional contribution of SOS1 and/or SOS2 to the process of CML development, here, we isolated the compartment of hematopoietic stem cells (HSPCs) from p210^BCR/ABL^ transgenic animals of the four relevant SOS genotypes (WT, SOS1-KO, SOS2-KO, and SOS1/2-DKO) and compared their functionality using in vitro colony-formation assays under different experimental conditions, including a TAM treatment for SOS1 ablation and/or imatinib treatments (Figure 5).

Our CFU assays using bone-marrow-derived cells from mice of the four relevant SOS genotypes carrying the p210^BCR/ABL^ transgene showed that the loss of SOS1 (p210^BCR/ABL^/SOS1^fl/fl^ mice treated with TAM) causes a drop in comparison to the WT controls in the total number of growing CFU colonies generated in these assays (Figure 5A,B and Appendix A). In contrast to the behavior of HSPCs from transgenic SOS1KO mice, the ablation of SOS2 (as well as the imatinib treatment in all the different genotypes) did not decrease the total number of colonies (Figure 5A) but significantly reduced the size (Figure 5C) of the colonies generated in our CFU assays in comparison with the control WT or SOS1^fl/fl^ mice. Indeed, these observations of the single SOS1-KO or SOS2-KO mice are consistent with the behavior shown by the HSPCs from transgenic SOS1/2-DKO cells, where both the colony number as well as the colony size were found to be significantly reduced in comparison with the rest of the genotypes (Figure 5A–D).

The differences in the number, growth rate, and size of the HSPC colonies arising from the SOS1-KO or SOS2-KO transgenic mice suggest that these two GEF isoforms act at different levels of stem cell renewal and proliferation and are also consistent with the prevalent role shown by SOS1 in our previous assays of CML development in these CML-prone mouse strains. 

## 4. Discussion

Using MEFs and BMDCs derived from SOS1-KO mice [17] we previously reported that the ABL-mediated phosphorylation of SOS1 promotes its RAC GEF activity and contributes to BCR-ABL-driven leukemogenesis. Specifically, in vitro assays in K562 cells showed that SOS1 is required for the BCR/ABL-mediated activation of RAC1, cell proliferation, and cellular transformation. Furthermore, using in vivo mouse xenograft assays, we also observed the significantly delayed development of leukemia upon the transplantation of BMDC cells expressing a retrovirally inserted p210^BCR/ABL^ construct [17]. Since, under normal physiological cellular conditions, SOS1 is known to be expressed in myeloid cells and is regulated by the ABL kinase in the processes of RTK-induced actin cytoskeleton remodeling [32], these observations of SOS1/2-KO mice provided substantial indirect experimental evidence suggesting that SOS1 may be a critical mechanistic contributor to the pathogenesis of CML.

In an effort to produce further direct experimental evidence supporting the above hypothesis, in this report, we have evaluated the effect of specific SOS1 and/or SOS2 genetic ablation on a mouse model of p210^BCR/ABL^-driven CML. For this purpose, we generated an ad hoc mouse colony resulting from cross-mating a known transgenic p210^BCR/ABL^-driven CML mouse model [33,34] with our SOS1/2-KO system [19,26,35] (allowing us to generate WT, SOS1-KO, SOS2-KO, and SOS1/2-DKO mice). Using this combined {Tec-p210^BCR/ABL^|SOS1/2-KO} genetic system, we investigated the phenotypic and functional effects of direct SOS1 and/or SOS2 ablation on the initiation/progression of CML in adult mice. Overall, our observations showed that SOS1 genetic ablation caused a specific suppression of various phenotypic manifestations of CML development in this mouse model.

Our analysis of the survival curves of transgenic p210^BCR/ABL^ mice of different SOS1/2 genotypes revealed that starting the TAM induction of SOS1 ablation at 6 months of age, when the pathological hallmarks of CML are not yet manifested in the transgenic mouse strain used in these studies [33,34], resulted in a significant extension of lifespan and survival in comparison with the untreated control mice. In contrast, in parallel studies, the constitutive ablation (from birth) of SOS2 showed a much more modest improvement in survival (about 50% lower) relative to the control WT transgenic mice. These observations suggest a significant contribution of SOS1 to CML development and support the notion that, as it happens with various other cellular functionalities previously characterized in our single and double SOS1/2-KO strains [19,26,27,31,35], SOS2 may just play a partially overlapping or ancillary role relative to SOS1 with regard to the development of CML. Interestingly, SOS1 ablation induced at a later age (8 months), when the onset of the disease has typically already started in these transgenic mice [33,34], did not produce any significant improvement in survival as compared to the control mice, suggesting a predominant functional contribution of SOS1 to the early stages of initiation, rather than to the progression of CML disease. Consistent with these views, we also observed that SOS1 ablation (but not SOS2 ablation) caused a very dramatic reversal of various typical hematological CML-related pathological alterations [33] involving different peripheral blood cell populations as well as internal organs relevant for hematopoiesis, such as the spleen or the liver. Finally, our characterization of the population of stem cells present in the bone marrow of our transgenic mouse strains lacking SOS1 or SOS2 was consistent with our previous observations in non-CML contexts [35] and provided further mechanistic insight regarding the participation of SOS1 in CML pathogenesis by documenting the prevalent functional role of SOS1 in the control of cell renewal and proliferation of this population of hemopoietic progenitor cells, which are responsible for giving rise to the various blood cell lineages that are pathologically altered in CML. In summary, all these observations in our experimental mouse system provide direct experimental evidence demonstrating the critical requirement of SOS1 function for the development/pathogenesis of CML in mice, thus validating the consideration of SOS GEFs as potential therapeutic targets for the treatment of this disease in human patients.

Our present observations uncovering the critical requirement of SOS1 for CML pathogenesis in mice are also highly consistent with and confirm previous independent reports on different aspects of human CML, also pointing to the relevant role this particular RAS-GEF plays in leukemogenesis. In this regard, the genomic landscape of acute myeloid leukemias is known to show frequent alteration of components of RAS pathways [45], where SOS1 is a recognized key regulator of downstream signaling initiated not only by normal, non-mutated RAS cellular proteins but also by ontogenically mutated, cancer-inducing RAS isoforms [18,46]. Remarkably, a unique dependence on SOS1 in KRAS^G12D^-induced leukemogenesis has also been previously reported based on the detection of SOS1 overexpression in KRAS^G12D/+^ cells and the observation that SOS1 deletion ameliorates oncogenic, KRAS-induced myeloproliferative neoplasm (MPN) phenotypes and prolongs the survival of KRAS^G12D/+^ mice [47]. Furthermore, our unsupervised computational analysis of the genomic data available in the CRISPR dataset of the Cancer Dependency Map portal [48,49,50] for multiple cancer and blood lineage cell lines, has assigned SOS1 (but not SOS2) the highest dependency score (dp score = −1.26) of CML cell lines in the database (Figure 6). Interestingly, the second-highest dependency score corresponded to the subset of cell lines encompassing AML, another relevant myeloproliferative disorder (dp score = −0.54), whereas no significant dependency scores were calculated for other myeloproliferative disorders such as CLL or ALL. On the other hand, SOS2 did not show any specific dependency on any specific type of cancer and only showed a very minor deviation (dp score = −0.08) of its dependency score in the case of the CML subset of cell lines (Figure 6). Indeed, all these analyses of human CML genomic samples confirm the critical role played by SOS1 in CML pathogenesis and support its consideration as a novel and additional molecular target for the design of future therapy approaches against this disease.

In this regard, our previous studies characterizing SOS1/2-deficient cells have demonstrated the functional prevalence of SOS1 over SOS2 with regard to cellular proliferation and viability [26,29,30,35] and, in particular, a specific critical requirement of SOS1 RAS-GEF function for mitochondrial dynamics, metabolism, and redox homeostasis [19]. Consistent with our observations, and regarding potential new therapeutic approaches to CML, some recent reports have described an increased sensitivity of AML leukemia to mitotoxic drugs or that targeting mitochondrial OXPHOX eradicates therapy-resistant CML stem cells [51,52]. In particular, a recent report indicated that targeting SOS1 overcomes imatinib resistance in CML [53,54], raising interest in clinical testing against CMLs for various recently developed small-molecular antiSOS1 drugs [28,55,56,57,58,59], which could prove particularly useful as an alternative when resistances frequently arise after sustained treatment with the classical imatinib or other second-generation TKI inhibitors [8,60].

## 5. Conclusions

Our previous in vitro studies using MEFs and BMDC derived from SOS1-KO mice provided indirect evidence suggesting the participation of SOS1 in the pathogenesis of CML. Using a combined {Tec-p210^BCR/ABL^ | SOS1/2-KO} mouse genetic model specifically generated for this report, we have now produced direct in vivo experimental evidence indicating that SOS1 genetic ablation causes the specific suppression of a variety of hematological alterations that constitute the typical hallmarks of p210-driven CML in mice and also significantly reduces the colony-forming ability of the population of HPCS from the bone marrow of p210^BCR/ABL^ transgenic mice.

Our observations demonstrate that the functional contribution of SOS1 is critically required for CML pathogenesis and support the consideration of this particular cellular GEF as a novel and alternative therapeutic target for CML, which might be particularly useful in cases where drug resistances frequently arise after sustained clinical treatment with imatinib or other TKI inhibitors.

## Figures and Tables

**Figure 1 cancers-14-03893-f001:**
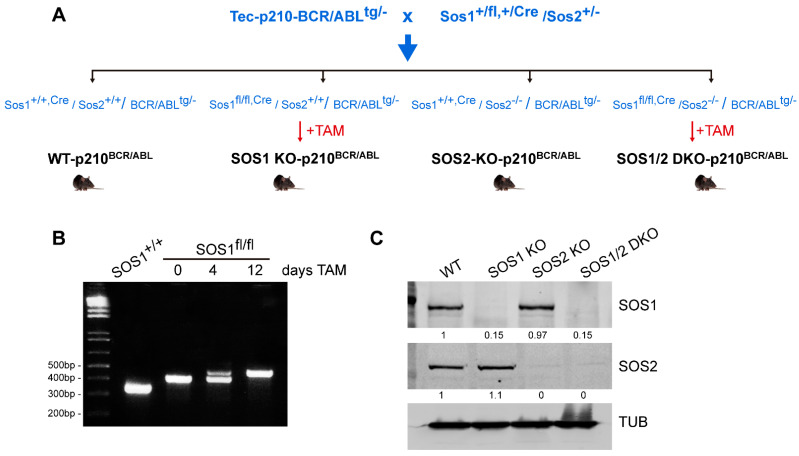
Generation of the {Tec-p210^BCR/ABL^|SOS1/2-KO} mouse model. (**A**) The transgenic mouse strain expressing p210^BCR/ABL^ under the control of the mouse Tec promoter [33,34] was cross-mated to our TAM-inducible SOS1/2-KO [19,26,35] mice in order to allow the generation of p210^BCR/ABL^-expressing mice of the four relevant SOS genotypes (WT, SOS1-KO, SOS2-KO, and SOS1/2-DKO), as indicated. SOS1-KO and SOS1/2-DKO were obtained after 12 days of tamoxifen diet, while SOS2 is a constitutive KO. (**B**) PCR amplification of DNA extracted from the tails of the control and SOS1^fl-Cre/fl-Cre^ mice. Partial excision of a SOS1 exon (10) in the CD25H domain was already visible at day +4, and complete exon removal was accomplished after 12 days of TAM treatment. (**C**) Western blot showing SOS1 or SOS2 protein depletion in isolated bone marrow cells from p210^BCR/ABL^-expressing, WT, SOS1-KO, SOS2-KO, and SOS1/2-DKO mice after 12 days of TAM treatment. Tub: Tubulin. The uncropped blots are shown in Appendix A.

**Figure 2 cancers-14-03893-f002:**
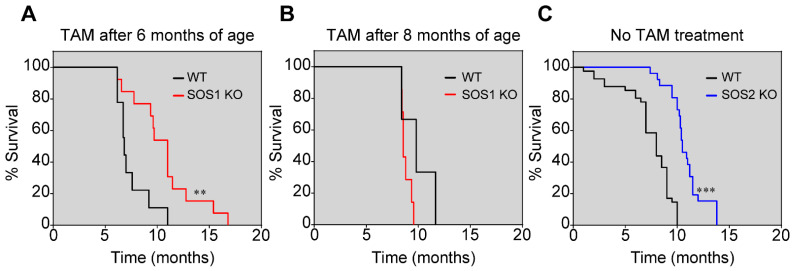
Increased survival of p210^BCR/ABL^ transgenic mice upon SOS1/2 ablation. Kaplan–Meier survival plots show that single SOS1-KO (red lines) or SOS2-KO (blue lines) transgenic mice increase their survival in comparison with transgenic SOS1 and 2-WT mice (black lines). (**A**) Animals were fed with tamoxifen-containing chow from the age of 6 months to induce the ablation of SOS1. WT: n = 9; SOS1-KO: n = 13; ** *p* < 0.01 vs. WT. (**B**) Animals were fed with tamoxifen-containing chow from the age of 8 months to induce the deletion of SOS1. WT: n = 3; SOS1-KO: n = 7. (**C**) Animals without tamoxifen treatment. WT: n = 8; SOS2-KO: n = 19; *** *p* < 0.001 vs. WT.

**Figure 3 cancers-14-03893-f003:**
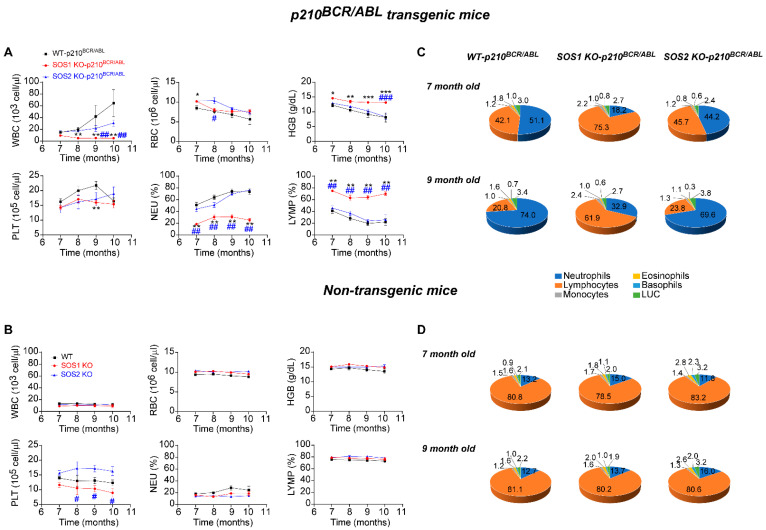
Suppression of CML hematological hallmark alterations in SOS1-KO/p210^BCR/ABL^ transgenic mice. (**A**,**B**) Kinetics of hematological parameters from peripheral blood of p210^BCR/ABL^ transgenic (**A**) and non-transgenic (**B**) mice of the indicated, color-coded SOS genotypes (WT, SOS1-KO, and SOS2-KO). The animals were treated with tamoxifen from 6 months of age to achieve inducible SOS1 ablation. The plots represent the total number of white blood cells (WBC), red blood cells (RBC), platelets (PLT), hemoglobin content (HGB), the percentage of neutrophils (NEU), and lymphocytes (LYMP) in the peripheral blood of p210^BCR/ABL^ transgenic mice (**A**) and non-transgenic mice (**B**). Values represented are mean ± S.E.M. n ≥ 12 for each genotype. * vs. WT mice, # vs. SOS2-KO mice; **, ## *p* < 0.01; ***, ### *p* < 0.001. (**C**,**D**) Percentage distribution of granulocytic and lymphocytic populations in the peripheral blood of p210^BCR/ABL^ transgenic (**C**) and non-transgenic (**D**) mice of the indicated SOS genotypes (WT, SOS1-KO, and SOS2-KO) at 7 months of age and 9 months of age, as indicated. The pie charts represent the percentage of neutrophils, lymphocytes, monocytes, eosinophils, basophils, and large unstained cells (LUC) (color-coded as indicated) in p210^BCR/ABL^ transgenic mice (**C**) and control non-transgenic mice (**D**). Values represented are the mean ± S.E.M. n ≥ 12 for each genotype. * vs. WT mice, # vs. SOS2-KO mice, *, # *p* < 0.05; **, ## *p* < 0.01; ***, ### *p* < 0.001.

**Figure 4 cancers-14-03893-f004:**
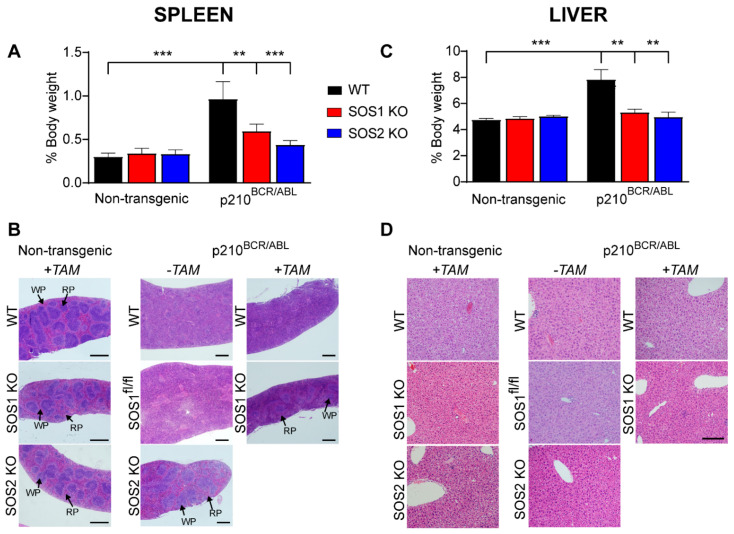
Suppression of CML-associated spleen and liver alterations in SOS1/2-KO/p210^BCR/ABL^ transgenic mice. Analysis of the spleen (**A**,**B**) and the liver (**C**,**D**) of 9-month-old p210^BCR/ABL^ transgenic and non-transgenic mice. (**A**,**C**) Bar plots representing the percentage of total body weight corresponding to the spleen (**A**) and the liver (**C**) of control, non-transgenic, and p210^BCR/ABL^ transgenic mice of the indicated (color coded) SOS genotypes (WT, SOS1-KO, and SOS2-KO). Values represented are the mean ± S.E.M. n ≥ 6 for each genotype). **, *p* < 0.01; ***, *p* < 0.001. (**B**,**D**) Hematoxylin and eosin (H&E) staining of histological sections of the spleen (**B**) and the liver (**D**) of 9-month-old non-transgenic and p210^BCR/ABL^ transgenic mice of the indicated genotypes. Where indicated, animals were fed with TAM-containing chow from the age of 6 months to induce the ablation of SOS1. Spleen red pulp (RP) and white pulp (WP) regions are indicated by arrows. Spleen scale bar: 500 μm; liver scale bar: 50 μm.

**Figure 5 cancers-14-03893-f005:**
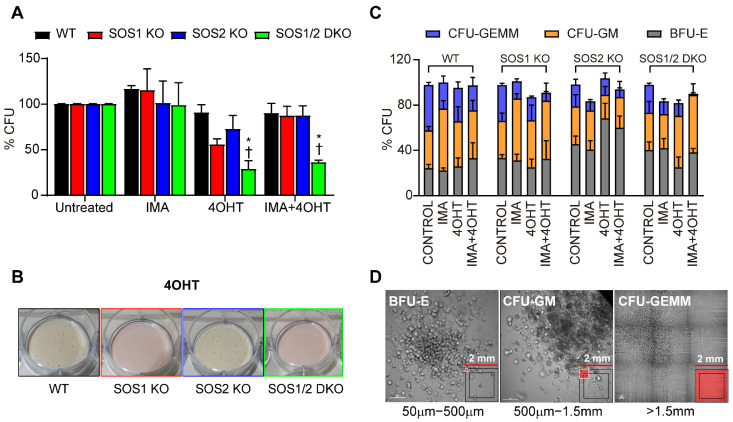
Analysis of the hematopoietic stem cell population from p210^BCR/ABL^ transgenic mice of different SOS genotypes. (**A**) Relative numbers of colony-forming units (CFU) generated by the CD117^+^ cells isolated from the bone marrow of 9-month-old p210^BCR/ABL^ mice of the indicated SOS genotypes (WT, SOS1-KO, SOS-KO, and SOS1/2-DKO) after treatment of the cultures for 12 days with 4 hydroxy-tamoxifen (4OHT, 2 μM), imatinib (IMA, 3 μM), or both (4OHT + IMA), as indicated. (**B**) Representative images of CFUs generated by CD117 + cells from the bone marrow of mice of the indicated SOS genotypes cultured for 12 days with 4OHT. (**C**) Percentage of different subtypes of committed progenitors from the bone marrow of p210^BCR/ABL^ transgenic mice of the indicated SOS genotypes after treatment in cultures with the indicated compounds for 12 days. Colonies were quantified according to their morphology: CFU-GM, colony-forming unit-granulocyte/macrophage; BFU-E, colony-forming unit-erythroid/burst-forming unit-erythroid; CFU-GEMM, colony-forming unit-granulocyte/erythroid/macrophage/megakaryocyte. (**D**) Representative images showing the morphology of the different CFU subtypes from WT untreated (control) cells following 12 days of culturing. Values represented in (**A**,**C**) are the mean ± S.E.M. n ≥ 3 for each genotype. * vs. WT; † vs. untreated; *, † *p* < 0.05.

**Figure 6 cancers-14-03893-f006:**
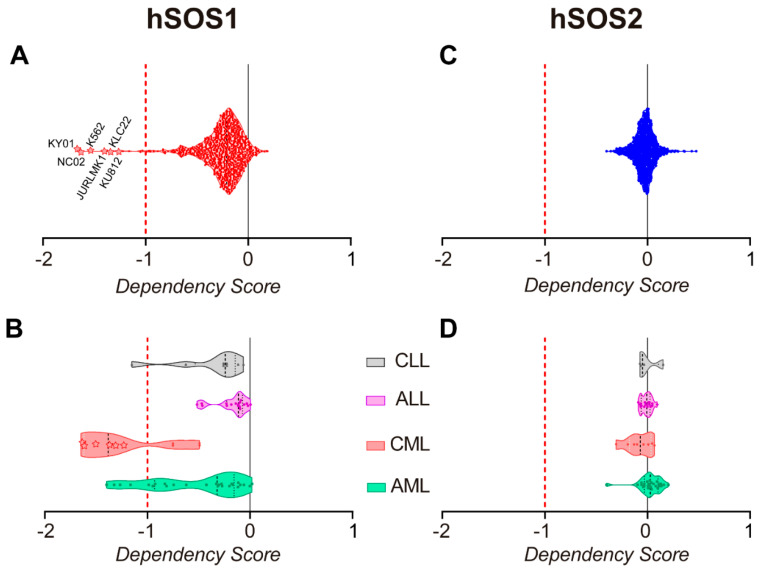
hSOS1 and hSOS2 gene dependency score in human cell lines from myeloproliferative disorders. (**A**,**C**) Plots of hSOS1 (**A**) and hSOS2 (**C**) dependency score values (*dp*) calculated for 1087 human cell lines corresponding to all different types of human cancers, which are gathered in the CRISPR library of the DepMAP portal (DepMap 22Q2 Public + Score, https://depmap.org/portal/, accessed on 22 June 2022). Dependency score values lower than −1 indicate that a given cell line is highly dependent on that particular gene for proliferation or survival. Notice that the most highly hSos1-dependent cell lines are CML cell lines (KY01, NC02, K562, JURLMK1, KCL22, KU812), and their *dp* values (*dp* < −1) are identified with a star. (**B**,**D**) Plots of hSOS1 (**C**) and hSOS2 (**D**) dependency score values (*dp*) calculated for the subset of human cell lines from the CRISPR library of the DepMap portal (n = 60) encompassing malignancies originating from different blood cell type lineages. Chronic lymphocytic leukemia (CLL), acute lymphocytic leukemia (ALL), chronic myeloid leukemia (CML), and acute myeloid leukemia (AML). The stars in panel B identify the same CML cell lines recognized by name in panel A.

## Data Availability

Data supporting the reported results can be obtained from the corresponding author.

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
