# Peer review of "Critical Requirement of SOS1 for Development of BCR/ABL-Driven Chronic Myelogenous Leukemia"

_cancers, 2022, doi:10.3390/cancers14163893_

Round 1

Reviewer 1 Report

In the present manuscript "Critical Requirement of SOS1 for Development of BCR/ABL- 2 Driven Chronic Myelogenous Leukemia" by Gomez et al, the role of the two genes SOS1 and SOS2 in the development of CML is discussed. The authors use a very sophisticated system of conditional, constitutive and transgenic alleles, whose time-consuming crossing together is often* not sufficiently appreciated by many researchers. The authors show that SOS1 in particular is involved in CML leukemogenesis and that pharmacological inhibition of SOS proteins may represent a novel therapeutic strategy against refractory or treatment-resistant CML. In principle, I find this study highly interesting and support publication in Cancers. Nevertheless, I have some comments on the manuscript that should be addressed before publication:   Line 111: could you please briefly provide some information about the system used. Which exon is floxed in the SOS2fl/fl mouse? Which exon is/which exons have been deleted in the SOS2-/- mouse? Which promoter is used for the CreERtg mouse? Is a CreERT2 mouse or a CreER mouse used?   Line 165: This must be the FLT3 ligand and not FLT3.   Line 231: I doubt the values given here for increased lifespan extension for SOS1 over SOS2 because the wt controls in these two experiments are also somewhat different. I think you need a much higher number of mice to make this statement. Also, the SOS2 KO seems to increase longevity more significantly (3 stars vs 2 stars). Additionally, I am missing the control here that the SOS1 / SOS2 single knockouts do not cause a phenotype without the P210BCR/ABL transgene, which has to be shown.    Figure 3 A/C: According to your Kaplan-Meyer curves, the SOS1 and SOS2 KO mice develop leukemia at a later time point. Do the resulting leukemias show identical phenotypes as in the WT mice? Or is it possible that a completely different phenotype develops here?   Figure 4 A: What do the spleens of SOS1 and SOS2-KO mice look like at about 13 to 15 months of age after the onset of the disease? Does a typical CML develop? Or does a different phenotype develop?   Figure 5A: Here, I miss the absolute numbers of CFUs, as 9-month-old P210BCR/ABL-expressing mice should have an increased number of CFUs compared with SOS KO mice. These absolute values should be reported in a separate figure (possibly in the supplements).    Line 418: As already written above, I believe this statement to be very risky and not statistically supported.     Line 486: You must mean OXPHOS and not OXFOX.    Line 489: Your results suggest that antiSOS1 drugs may help in the development of CML, but according to your results they would have to be taken during leukemogenesis, as later treatment (Fig. 2B) seems to have no benefit anymore. Possibly, however, a pan-SOS inhibitor would be interesting.  

Author Response

We thank this reviewer for her/his overall positive appraisal of our work indicating, in particular, that “…The authors use a very sophisticated system….”, or that “….I find this study highly interesting and support publication in Cancers “.

Our response to the specific comments made by the reviewer is included in the attached file. Please see the attachment

Reviewer 2 Report

The paper reports an interesting finding consisting on the implication of SOS1 in the progression of CML. It is well known that most CMLs arise from the fusion of BCR to the ABL kinase, but little is known about the downstream pathways that lead to the development of CML. Although fortunately in most cases this type of leukaemia can be initially controlled with the ABL kinase inhibitor imatinib (Glivec), very often CML develops resistance to imatinib and progresses. So it is important to describe downstream pathways to find new pharmacologic targets. The authors use transgenic mice expressing the BCR-ABL fusion that develops CML after more or less half a year. These mice are crossed to SOS1 and SOS2 KO strains to study the relevance of these two adapters on CML development. Notably, they circumvent the problem of early lethality of SOS1 KO by using a cre-lox system that can be activated with tamoxifen, so they can supress SOS1 just before the CML is supposed to develop. Using this system, the authors demonstrate that SOS1 but not SOS2 is relevant for BCR-ABL induced CML. In my opinion, the paper is interesting well written and the conclusions are clear, however there are several questions that should  be addressed before the work can be published.

Fig3. A and B

The units in the Y axes of plots showing  red blood cells and platelets number are not correct taking into account that in mice the normal values per microliter are between five and seven millions red blood cells and around half a million of platelets.

Fig2 B.

In my opinion, the n of control group in this experiment is too low to make conclusions out of it. I understand it is a long experiment to repeat, however, you could compare the survival of SOX1 KO group that receive Tamoxifen at 6-month with the SOX1 KO group that received it at 8 month to reach more or less the same conclusion.

In line 371, the authors say “Our CFU assays using bone marrow-derived cells from mice of the 4 relevant SOS 371 genotypes carrying the p210BCR/ABL transgene) showed that the loss of SOS1 (p210BCR/ABL/SOS1fl/fl mice treated with TAM) causes a significant drop in comparison to the WT controls in the total number of growing CFU colonies generated in these assays (Figure 5A, B)”. This stamen is not correct, as it is now shown, in figure 5A the only situation in which the change is labelled as significant is the double SOS1/2 DKO, although it is true that the SOS1 KO decreases this percentage, the decrease is not indicated as significant. So, a significance label should be added to the bar graph or remove the term significant in the text.

In line 375, the authors say “In contrast to this behavior of the HSPCs from transgenic SOS1KO mice, the ablation of SOS2 (as well as the treatment with Imatinib of all different genotypes) did  not decrease the total number of colonies (Figure 5A) but significantly reduced the size 377 (Figure 5C)” I don’t find any size comparison in this figure. Yes, the percentages of each colony type (CFU-GEMM, CFU-GM and BFU-E) were evaluated by their shape and size, but no data reporting the size or the comparison of the colony size in each treatment is reported Moreover, colony identification by their appearance is a not very reliable method to identify them, in my opinion, it should be supported using immunologic markers.

Author Response

We thank this reviewer for her/his overall positive appraisal of our work indicating, in particular, that “The paper reports an interesting finding ….” or that “….I , the paper is interesting well written and the conclusions are clear…. “.

Our answers to the specific questions raised by this reviewer are included in the attached file. Please see the attachment

Reviewer 3 Report

Dear authors:

This is a continuous story of the published story (DOI: 0.1038/s41388-021-01886-3). It showed that the SOS1 is critical to CML (typically with the P210 fusion gene mutation). You used the mouse model of SOS1/2 Tamoxifen-knock out model for the research, and found that the SOS1 depletion is critical to the CML development in mice. The research results seem to be promising and agrees to the conclusion. 

First, the paper is not comprehensive to readers as many details were not referenced. Below are some examples:

1. Please add the reference of the mice model development so readers could understand why Tamoxifen is capable of knocking out the SOS1.

2. Please add reference to the figure 5 that why CD117+ cells were used as the hematopoietic stem cells. Also add reference that CFU% is an indication for CML development. 

3. Please explain why use the concentration of 3uM IMA to compare with the 4OHT of 2uM. Also, add the reason why using IMA for the assay (I believe it is a standard TKI treatment of CML and resistance can be developed along the treatment).

4. Please explain the function of 4OHT and why to use it in the figure 5 after SOS1 knock out.

5. The method of computational data (figure 6) was not described and it is not clearly interpreted to the readers that are not in the same field. It would be nice if a brief explanation is given. 

Then, however, I found that the paper's research topic has been studied and SOS1 has been noticed to be essential in CML development. You may find similar conclusions from paper 'Phosphorylation of SOS1 on Tyrosine 1196 promotes its RAC GEF activity and contributes to BCR-ABL Leukemogenesis' and paper 'Targeting SOS1 overcomes imatinib resistance with BCR-ABL independence through uptake transporter SLC22A4 in CML.' Instead of knock out SOS1, they did SOS1 knock down to the in vitro and in vivo models and noted with similar results. It is nice to see SOS1 knockout model in p210 mouse, but to publish in this journal, the paper is encouraged to show more in details why and how SOS1 is driving the leukemogenesis.

Author Response

We are grateful for the positive assessment of this reviewer indicating in particular that our research results “…seem to be promising and agrees to the conclusion” 

Our answers to the specific questions raised by this reviewer are included in the attached file. Please see the attachment 

Round 2

Reviewer 1 Report

The authors have answered all my comments to my full satisfaction. I would still like to ask the authors to not only mention the information about the SOS1 and SOS2 knockout mice, but also briefly mention to which Cre mouse line de floxed mice have been crossed. This is an essential part of the system and should also be mentioned. 

When this is inserted, this manuscript can be accepted for publication in Cancers.

Author Response

Our response to the comment of reviewer 1 involves a short text addition (“ …., where SOS1 ablation is controlled by CreERT2”) at the end of the sentence starting in line 111.  

Thank you very much for your consideration and help.

Reviewer 3 Report

Thanks for the response. The manuscript is much more clear in order and background description is also better. 

Author Response

Thank you very much for your consideration and help.